# Barriers and enablers to the effective implementation of robotic assisted surgery

**Louisa Lawrie**[1]*, **Katie Gillies**[1], **Eilidh Duncan**[1], **Loretta Davies**[2], **David Beard**[2], **Marion K. Campbell**[1]

1 Health Services Research Unit, Institute of Applied Health Sciences, School of Medicine, Medical Sciences and Nutrition, University of Aberdeen, Aberdeen, Scotland, United Kingdom, 2 RCS Surgical Interventional Trials Unit (SITU), Nuffield Dept Orthopaedics, Rheumatology and Musculo-skeletal Sciences, University of Oxford, Oxford, United Kingdom

* louisa.lawrie1@abdn.ac.uk

**Data Availability Statement:** All relevant data are within the manuscript and its Supporting Information files.

**Funding:** MKC and DB were funded via an unrestricted grant (https://www.intuitive.com/en-

## Abstract

### Background

Implementation of Robotic Assisted Surgery (RAS) is complex as it requires adjustments to associated physical infrastructure, but also changes to processes and behaviours. With the global objective of optimising and improving RAS implementation, this study aimed to: 1) Explore the barriers and enablers to RAS service adoption, incorporating an assessment of behavioural influences; 2) Provide an optimised plan for effective RAS implementation, with the incorporation of theory-informed implementation strategies that have been adapted to address the barriers/enablers that affect RAS service adoption.

### Methods

Semi-structured interviews were conducted with RAS personnel and stakeholders, including: surgeons, theatre staff, managers, industry representatives, and policy-makers/commissioners. The Theoretical Domains Framework (TDF) and the Consolidated Framework for Implementation Research (CFIR) was used to identify barriers and enablers that represent individual behaviours, capabilities, attitudes, beliefs, and external organisational factors that influence the implementation of RAS.

### Results

Findings suggest that implementation planning has three separate phases–pre-, early, and late implementation. For pre-implementation, barriers and enablers identified included the cost of RAS equipment and issues of economic viability, weak outcome evidence for RAS, a preponderance of an eminence driven model, the clinician/manager relationship, and views around the uptake and expansion of RAS in the future. Early implementation findings revealed role changes for theatre personnel and an enhanced team approach, reliance on industry for training provision, and changes in skill sets and attentional processes. Late implementation factors included equipment maintenance costs, technological limitations, changes to cognition during RAS routine use, and benefits to institutions/healthcare professionals (such as ergonomic improvement).

gb) awarded by Intuitive Surgical (European Research Board). The funder had no role in the conception, design, conduct, analysis, or interpretation of the study.

**Competing interests:** The authors have declared that no competing interests exist.

## Conclusion

Together, findings suggest the factors that affect RAS implementation are multi-faceted and change across the life-cycle of intervention adoption. Theory-informed strategies are suggested which can optimise implementation of RAS. Optimisation strategies need planning from the outset.

## Introduction

Robotic Assisted Surgery (RAS) is becoming increasingly prevalent worldwide [1]. Since its initial introduction, RAS platforms have evolved and currently vary in size, versatility, and clinical/specialty use [2]. Reports of RAS advantages have highlighted improved ergonomics, precision, tissue magnification, reduced reliance on surgical assistance as well as efficient surgical training and workflow practices [3–7]. Although the full impact on patient and clinical services is still being assessed, it seems inevitable that the adoption of robotic systems will increase rapidly over the next decade [2, 8]. The integration of RAS services into the wider clinical system is disruptive, requiring significant investment in terms of equipment costs, service alignment and workforce training. Successful implementation of RAS is particularly complex as it requires both major capital outlay and physical adjustment for the new technology, but also requires individuals and organisations to significantly change processes and behaviours to work with the new systems.

Whilst the major infrastructure impacts are increasingly being recognised, the barriers and enablers to efficient behaviour and process change that support effective implementation of RAS are less understood. Insights from the field of implementation science in the healthcare setting have shown that exploring behavioural influences can be a highly effective way to maximise the chances that an intervention is implemented quickly and effectively [9–13].

With the global objective of optimising and improving RAS implementation, the aims of this study were therefore, to: 1) Explore the barriers and enablers to the implementation/scale up of RAS services at different stages of the implementation process focusing on technology-specific influences, but also with an augmented focus on behavioural influences; 2) Provide an optimised implementation plan for RAS integration, with the incorporation of theory-informed implementation strategies that have been adapted to address the barriers/enablers that affect RAS service adoption.

## Methods

### Design

This study used semi-structured interviews with personnel involved in both soft-tissue (cavity based) and orthopaedic RAS and several specialty areas. An interview topic-guide was developed to identify possible barriers and enablers to implementation. To ensure relevant behavioural and organisational elements were appropriately assessed, the topic guide was informed by two frameworks—the Theoretical Domains Framework (TDF) [12], which considers behavioural characteristics of healthcare professionals/individuals, and the Consolidated Framework for Implementation Research (CFIR), which emphasises organisational factors that affect implementation [14].

### Participants

We purposively sampled key personnel involved in the introduction, delivery, and evaluation of RAS services. This included: Surgeons (Users/Non-users of RAS), wider theatre staff (Scrub

Nurses and Anaesthetists), National Health Service (NHS) Managers, RAS Industry representatives, as well as Policy-makers/Commissioners involved in the evaluation of surgical innovations. We aimed to interview clinical stakeholders from a range of hospital sites across the UK and include a range of views (e.g. proponents and opponents) and experiences of RAS (i.e. variations in specialty and duration of RAS experience). A pre-specified sample of 35 was included to ensure full representation of stakeholders and saturation of themes. We also judged the sufficiency of our sample size based on the principles outlined by Francis and colleagues [15].

## Data collection

We tested our guide with experienced RAS clinicians ($n$ = 2) who provided feedback on the suitability of our questions and suggested missing relevant topics that could be explored. LL (female, academic researcher, PhD) conducted all of the interviews. Participants were not known to the interviewer. All members of the research team (MC, DB, KG, LL, LD) were present for the pilot interviews. An initial set of key informant interviews were conducted ($n$ = 5) with individuals known to have implemented RAS successfully. These key informant interviews were designed to ensure that we had a clear foundation and understanding of common implementation barriers and enablers. LL led the interviews. Other members of the research team (MC, DB, KG, LD) were also present for interviews at this stage. Participants with a diverse range of experiences were then invited to interview via email. These interviews were conducted and led by LL and supported by DB or MC.

Recruitment targeted RAS personnel who were identified via the Royal College of Surgeons (RCS) Robotic and Digital Surgery Initiative (RADAR) network of contacts. All specialties who had RCS appointed leads and extensive professional networks were asked to nominate other potential interviewees with relevant RAS experience. Social media was also used to advertise the study, as well as existing contacts of individuals with relevant experience. Data collection took place between October 2020 –March 2021. All participants were provided with an information leaflet in advance of the interview. Field notes were generated after the interviews to note key information. All interviews were conducted via Microsoft Teams [16]. Audio-recordings of interviews were transcribed verbatim, anonymised, and subjected to the analysis process outlined below.

## Data analysis

A coding guide was used to aid interpretation of the transcripts (informed by the TDF and CFIR): this was developed (LL), verified (ED, KG) and updated iteratively during analysis. Three of the 35 interview transcripts were double coded independently (LL, ED, KG) prior to comparing the coding results. Any coding discrepancies identified during this process were discussed to reach consensus. One researcher (LL) coded all interview responses into the relevant theoretical domains. NVivo 11 [17] was utilised to facilitate data analysis. Core domain barriers and enablers were identified based on reported frequency, the presence of conflicting opinions and/or the extent to which a domain barrier/enabler was thought to impact implementation [13]. Relevant CFIR domains and respective constructs were assessed based on similar criteria. Themes were mapped to three main phases of implementation (pre-implementation, early implementation, and late implementation). This analysis provided a behavioural diagnosis of the primary factors influencing effective implementation.

**Developing bespoke theory-informed implementation strategies.** The findings from the interviews were used to develop a RAS implementation plan to address the key barriers identified at each phase. To do this, core themes raised by the interviewees were reviewed and

the actions repeatedly observed by interviewees to improve implementation were summarised. Additionally, we summarised the barriers and enablers identified in the interviews and matched evidence-based strategies to identified behavioural influences, a systematic method proposed by Michie and colleagues [18] shown to be highly effective in other clinical applications [9–12]. This approach makes use of a taxonomy of strategies that have been theoretically and experimentally proven to change specific behavioural targets (tailored to the context in hand, i.e. implementation of RAS). We have included more detail about the methodological steps involved in the design of the implementation plan in a supporting file (S1 File).

The suggested implementation strategies were collectively reviewed and refined by the research team. A RAS Surgeon reviewed the implementation plan to establish the feasibility of our proposals.

### Ethical approval

This study was approved by the College Ethics Review Board at the University of Aberdeen (CERB/2020/7/1984). Informed verbal consent was obtained from all participants.

## Results

### Participant demographics

Sample demographics are provided in Table 1 (age, gender, and ethnicity was not recorded for 1 participant). Twenty-two clinical interviewees including Surgeons, wider theatre staff and Service Managers across a range of specialties were sampled from a total of 16 hospital sites located across England, Scotland, and Wales. The majority of clinical interviewees worked in the NHS. Three clinical interviewees were from international institutions at the time of interview: 2 had previous experience of clinical practice in the UK. Interviews lasted an average of 55 minutes (ranged from 33 minutes– 1 hour, 20 minutes).

### Findings

It was apparent early on in the interviews that the types of barriers and enablers varied significantly depending on the stage of the RAS implementation lifecycle, with different barriers/enablers exerting major influences during three distinct phases of implementation:

1. Pre-implementation stage—decision to procure, up to the delivery of RAS equipment.

2. Early implementation stage—initial integration of a RAS service, after procurement of the equipment.

3. Late implementation stage—later phases of RAS adoption, during long-term integration of a RAS service.

As such, the findings summarise the reported barriers and enablers of RAS implementation for each of the three key stages. In each section, we summarise the core barriers/enablers identified and highlight (in brackets) the associated dominant theoretical domains.

A summary of the barriers and enablers at each stage of implementation is also depicted in Fig 1. This illustrates that some higher-level concepts remain consistent across the entire implementation lifecycle (e.g. environmental context, skills, and social influences), others less so. For example, the influence of professional role/identify is particularly exerted during pre-implementation, whereas issues such as the importance of "reinforcement"—the positive influence from reward and recognition—emerges strongly during pre-implementation for the organisation but in late-implementation for the clinical professionals.

**Table 1. Participant demographics.** Note that age, gender and ethnicity was not reported by one participant.

| Characteristic | N = 35 | % |
|---|---|---|
| **Age** | | |
| Median | 50 | |
| Range | 30–70 | |
| **Gender** | | |
| Female | 7 | 20.6 |
| Male | 27 | 79.4 |
| **Ethnicity** | | |
| Asian British | 3 | 8.8 |
| Caucasian | 29 | 85.3 |
| Indian | 1 | 2.9 |
| Other white background | 1 | 2.9 |
| **Role** | | |
| National Surgical Specialty Lead | 5 | 14.3 |
| Surgeon (RAS user) | 11 | 31.4 |
| Surgeon (Non-RAS user) | 2 | 5.7 |
| Scrub Nurse | 2 | 5.7 |
| Industry representative | 5 | 14.3 |
| Policy-maker/Commissioner | 5 | 14.3 |
| Surgical Trainee | 2 | 5.7 |
| Anaesthetist | 1 | 2.9 |
| Service Manager | 2 | 5.7 |
| **Specialty** (Specialty Leads, Surgeons & Trainees) | **N = 20** | **%** |
| Urology | 5 | 25.0 |
| Colorectal | 8 | 40.0 |
| General | 2 | 10.0 |
| Orthopaedics | 3 | 15.0 |
| Gynaecology | 1 | 5.0 |
| Thoracic | 1 | 5.0 |

## Pre-implementation barriers and enablers

Identified influences (positive and negative) when laying the foundations for the initial procurement and implementation of RAS are summarised below.

**Cost & culture (CFIR cost, culture).** Cost analysis issues are contained within all three phases of the study findings and are described in detail in the section on Outcomes of RAS & Influence on Adoption below.

The culture of a hospital and/or the NHS, having an appetite (or not) for innovation and a level of openness to the incorporation of new interventions were also identified as opportunities and challenges to implementing RAS. A lack of appetite and agility to respond to innovation was problematic:

> "Well what didn't help was the scepticism and the lack of agility. . ..and vision of big administrative structures and managerial structures in the NHS, where innovation and change is not really adopted very easily." P01S, Specialty Lead.

The transition of an innovation with pedigree and genesis seated in a more commercial or private healthcare delivery sector into a public medicine domain was further framed as a barrier according to some interviewees:

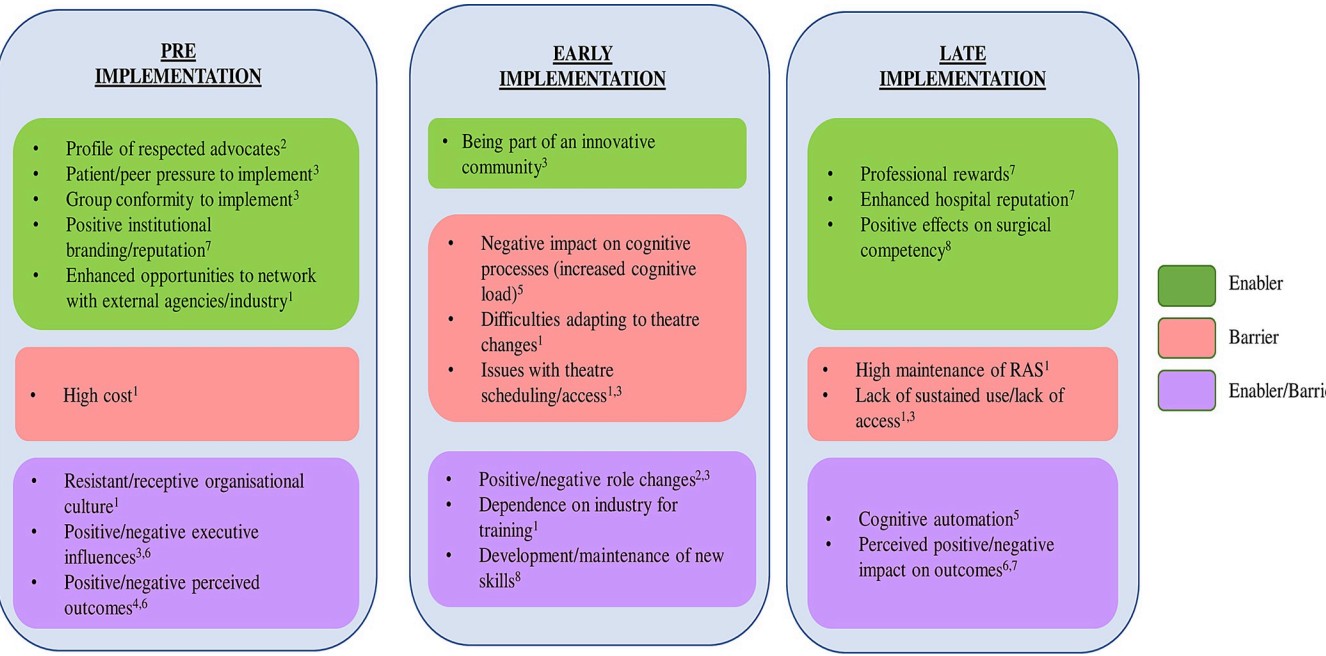

**Fig 1. The barriers and enablers of implementation at each phase of adoption.** TDF domains: [1]Environmental Context & Resources, [2]Social Professional Role & Identity, [3]Social Influences, [4]Beliefs about Consequences, [5]Memory Attention & Decision Processes, [6]Knowledge, [7]Reinforcement, [8]Skills.

"Trying to roll that process into a public healthcare system–and it's not the UK specifically, this is a Europe-wide experience–is where we seem to hit friction. As a company, we really struggle to work out what it is that means that that model doesn't necessarily translate to a public healthcare system." P03I, Industry stakeholder.

**Opportunity to network with external organisations/industry (CFIR cosmopolitanism and industry enabled opportunities).** For initial adoption, many interviewees cited positive experiences with industry in the process of procurement, namely in terms of effective collaboration with the sales team and establishing international and national links with external organisations. Exposure to other institutions around the world and observing the demonstrable effects of RAS created a favourable impression for one Service Manager:

"My understanding of the benefit of delivery of that was certainly going to [country] and seeing the system . . .there. . . . . It's a conveyor belt of surgery. . ..using robotics." P2M9, Service Manager.

**Social and professional roles: Impact on adoption (Dominant domains: Social influences, social professional role & identity).** Some Service Managers interviewed had a clinical background and/or worked in prior services where technology and innovation had been at the forefront: this was reported to contribute to their positive impression of RAS. Other interviewees acknowledged that their own established international, national and/or local status facilitated the adoption of RAS, sometimes with reference to their academic credentials. One industry representative considered the interplay between, and influence of academia and clinical Key Opinion Leaders (KOLs) on the slow change in perception into the more general workforce. On the one hand clinical KOLs, without necessarily utilising academic or research based arguments, had driven implementation by providing a positive impression. However,

those from more research-based academic backgrounds were also influential during pre-adoption, but in a more balanced and dispassionate way (offering arguments for and against implementation). Both of these types of individuals were considered to influence the next level of the RAS user workforce:

". . .But when you get into the next realm, when you get into that earlier doctor group who come out of academic institutions who are good surgeons and they start to say, "This we think is good", that's massively influential." P2I13, Industry stakeholder.

As a further nod to the relationship between academia and social/professional status, a few interviewees recognised that innovation in surgery, including RAS, generally tends to be 'eminence driven' as opposed to 'evidence driven'.

**Perceived peer/patient influence and group conformity (Dominant domain: Social influences).** Group conformity, peer and patient influences were also noted as drivers of RAS implementation. Observing the conduct of RAS in other hospitals was sometimes reported to drive local decisions to implement. Perceived patient pressure was also reported to influence procurement, especially in Urology where robotic prostate procedures are well-established:

". . .I think particularly with urology, you would see it with the prostate operation, that suddenly these centres appeared and every patient went where there was a robot." P2I26, Industry stakeholder.

While many interviewees reported a general enthusiasm about RAS among peers in the surgical community, some level of resistance and/or disengagement about adoption was commonplace. According to one stakeholder, competitiveness amongst clinicians from multiple disciplines within the same hospital could act as a barrier to initial RAS implementation; this was thought to be fuelled by the desire to incorporate other innovative technologies. Furthermore, while many interviewees reported a general enthusiasm about RAS among peers in the surgical community, it was acknowledged that there had been some level of resistance and/or disengagement about adoption:

". . .there is a lot of resistance I think to robotic surgery and I know because it's a very young specialty because it's only twenty years old, unlike laparoscopic surgery ten years prior to that. . .people I work with that are very resistant to it. . ..." P12N, Scrub Nurse.

**Managerial, hospital level executives and commissioner influences (Dominant domain: Social influences, related to knowledge).** Many interviewees reported difficulties in explaining the perceived benefits of RAS to management, hospital level executives and commissioners, especially in the absence of strong evidence and considering the up-front cost of the equipment:

"a key part of the actual procurement. . ..and convincing some people that this was a pursuit worth undertaking. . .and standing that against clearly the considerable costs involved." P2S22, Surgeon (RAS User).

This inability to articulate the benefits of RAS to management was sometimes a significant barrier to adoption, as the individuals involved in the management and executive structures of the organisation were instrumental in enabling implementation. One Specialty Lead

highlighted the essential requirement of convincing commissioners to overlook the stereotypical portrayal of surgeons and instead appreciate RAS benefits in isolation of any prejudicial views:

> "... to convince commissioners that, "Look, there is benefit here." Benefit is that it's not just a toy for the boys, which is how we are described very often..". P01S, Specialty Lead.

The perceived weakness of the evidence base and procurement structure could also be problematic in the process of obtaining a reasoned or rational procurement decision. The relative dependence on industry's value proposition and marketing information in the process was considered challenging. In addition, it can be easy for surgeons to persuade managers of the benefits of RAS, particularly if managers do not possess a sound understanding of the evidence.

> "... and the problem with the managers is they don't understand the evidence, so the surgeons can persuade them, I think, that there is evidence when there isn't." P2S25, Policy-maker/Commissioner.

**Outcomes of RAS & influence on adoption (Dominant domain: Beliefs about consequences, linked to knowledge).** Outcomes of RAS appeared to be a substantial influencer in its procurement. The Managers in the current study were convinced that the outcomes of RAS are positive due to data reported in scientific literature and/or discussions with surgeons, which contributed to the decision to implement RAS locally. Beliefs about positive patient outcomes associated with RAS were reported by Managers, such as shorter length of hospital stay and improved recovery as compared to open surgery, while other interviewees indicated that the patient outcomes associated with RAS, as reported in the literature, were not dissimilar to the outcomes associated with laparoscopic surgery in certain specialties. The notion that more robust evidence of RAS benefits is required to justify the introduction of RAS in the NHS was suggested by a few interviewees.

Many interviewees spoke about the evidence of RAS benefits in relation to a specific specialty and/or procedure, and one stakeholder acknowledged that it was very difficult to judge the benefits of RAS based on the heterogeneity that exists in the field of robotics, including the literature:

> "I think it's very muddled... And I think it's an area where it's impossible just to take the evidence for one and apply it to another. That is part of the difficulty." P2M10, Policy maker/Commissioner.

Nevertheless, some interviewees possessed disproportionately strong beliefs about the benefits of RAS. There was also acknowledgments that contrasting perspectives among the surgical community can be a hindrance to widespread adoption.

**Beliefs about long-term effects of RAS and impact on adoption.** One of the most influential (positive) drivers of RAS service implementation were the beliefs associated with the perceived long-term effects of implementation. Some interviewees indicated that long-term implementation could reduce overall health inequalities by widening access to minimally invasive surgery and enabling operations to be conducted on more patients, with positive outcomes:

> "...And, actually, that was something that won me over as well..... That the frail, elderly, and the obese patients can be operated on using the robot more... I think with better outcomes..." P2SM20, Policy-maker/Commissioner.

Other beliefs about outcomes which motivated implementation included the notion that RAS implementation would enable research and foster collaboration between institutions. Some interviewees also expressed the belief that introducing RAS on a wider scale could positively transform the way in which clinical staff and patients interact with the healthcare system, including the idea that robotic platforms facilitate digital data collection which can translate to positive patient and societal outcomes. In addition, the belief that data generated by robotic platforms can be used to improve workforce training was also cited as a reason to implement RAS. Many interviewees also acknowledged that robotic surgery is a rapidly expanding field. Some suggested that local implementation would be necessary due to the rapid national/international escalation of RAS, sometimes with reference to the implications for workforce training. The potential of acquiring a robot to attract surgical talent and retain existing staff, especially in some regions of the UK where recruitment/retention is problematic, was often cited as a motivator for local implementation.

"...I know from personal experience, colleagues who have given up because their Trust just did not want to go down robotic and they just did not see themselves doing the same old, same old, same old. So having access to new technology, innovation, stimulating things, a reason to go to work, has been really, really important". P14S, Surgeon (RAS user).

**Pre-implementation consideration of cost and economic viability.**   The significant upfront cost of RAS equipment was reported as a barrier to initial adoption. Many interviewees also acknowledged at an early stage that RAS would not be cost-effective if the robotic platform was not utilised in an economically appropriate manner. Acknowledgments of the importance of high surgical volume and ensuring optimum use, i.e. 'being able to sweat the asset/robot', was present during pre-implementation:

"... and also because it wasn't so cost effective unless you really sweated the asset, that to me was the most persuasive argument..." P2SM20, Policy-maker/Commissioner.

Similarly, one Industry representative recognised that while there were demonstrable patient outcomes of RAS in certain niche specialities, such as Head and Neck, the application of RAS in a single specialty context may not be seen to be worthy of investment due to the expected low surgical volume. There was recognition that multi-specialty implementation can allow for increased volume and program efficiency.

**Institution branding and profiling associated with RAS procurement (reinforcement).**   Some interviewees postulated that the introduction of RAS provides individuals with the opportunity to brand their respective hospitals as 'innovative' and modern. This was sometimes cited as an incentive for NHS hospitals to introduce RAS:

"I think the biggest driver even in the NHS at the moment is a driver to promote your organisation through the introduction of new technology..." P2S21, Specialty Lead.

### Set-up & early implementation barriers and enablers

**RAS influenced role modifications (Dominant domains: Social professional role & identity; social influences).**   Interviewees reported shifts in professional roles and duties as a result of implementation during the early phase of adoption. Increased multi-specialty exposure and greater intraoperative responsibilities were reported for some nursing staff. This characteristic was expressed both as a barrier and enabler. The Scrub Nurses in the current study

regarded adaptations to their roles as positive overall. However, initial difficulties in assimilating to changes were reported by one Scrub Nurse, namely regarding the confidence to perform in theatre:

> "...You literally are stepping over a big fence between something that's never been allowed to be your role to, oh no, this is what you're doing now. It did take time." P2N5, Scrub Nurse.

Notably, two interviewees suggested that RAS can be sub-optimal from a general surgical training perspective, particularly outside specific RAS training and when utilising a single console robotic platform (note that dual console robotic platforms, a more recent development, will not generate this issue). The lack of requirement for the classic "surgical assistant" can deprive trainees of close operative field experience and peri-operative verbalised teaching.

The greater separation of operative experience/technique from duties of pre-operative/post-operative medical care was also highlighted for RAS surgery by trainees, as well as access to dual console equipment, which may have a bearing on confidence and trust within theatre. There was controversy over the impact of RAS on the Anaesthetist's role and their need for involvement in the procurement process. Some interviewees suggested that the Anaesthetist's role becomes more demanding as a result of RAS, warranting their wider engagement, while others suggested that the change to their role is less fundamental:

> "there's not a huge amount of difference in terms of the actual anaesthetics that are administered, but there is a little bit of difference in terms of how the patient would be positioned, how they're handled in the operating theatre...." P03I, Industry Representative.

A few interviewees also mentioned changes to the Surgeon's role as a result of introducing certain robotic platforms in theatre: RAS often removes the requirement for a single 'master' operator and necessitates greater reliance on the surgical team. Whilst such a change can be seen to have merit for a process that is very team dependent, sometimes this change was reported to create resistance amongst some Surgeons who enjoy operating in a more solo mode during open cases. An increased dependence on a team approach could also bring limitations and be a barrier. There is an idea that the successful conduct of RAS depends on effective teamwork, therefore a stable and consistent team is required. It was reported that RAS is often conducted with team members not always familiar with each other for every procedure:

> "We needed the same people and everyone knew to be clear on the team what their roles were and the surgeon found that beneficial. They want the same team in there. As time goes on you're not able always to implement that because you can't use the same people every time, so then you get someone new in and then it kind of goes backwards a little bit ..." P12N, Scrub Nurse.

**Reliance on industry & international collaboration (Dominant domain: Environmental context & resources).** The need to rely on industry or international mentorship for the early implementation of RAS training created both opportunities and challenges for some interviewees. The opportunities were related to the training package resource provided and the challenges included the transition from company led support ("hand holding") to confident independent use. The responsibilities (such as for proctoring) in the transition phase between learning and established practice is not well defined and an area for attention. Commercial input extends to technical support, but not clinical.

**Perception of competencies required to conduct RAS (Dominant domain: Skills).**
There were conflicting views around whether RAS demands the same set of skills as other minimally invasive techniques, and sometimes this was quoted to be dependent on the robotic platform in use and the type of procedure conducted:

> "There is a skill set that needs to be trained to understand how to use the articulation, how to use clutching systems in robots that wouldn't exist at all in laparoscopy." P2I26 Industry stakeholder.

Surgical adaptation and acquiring 'good screen skills' was regarded as essential, according to a few interviewees:

> "Robotic is ten times magnified, so you've got to realise and adapt . . . when you think you're moving one centimetre, it looks a lot more than that on a screen and things like that, so yeah. Just that whole skill-set that I didn't have before, so yeah, I had to learn all of that." P2N5, Scrub Nurse.

Enhanced motor (dexterity) skills were cited to be important for the effective conduct of RAS, not dissimilar to laparoscopic surgery, while efficient surgical decision-making remained a necessity. Many interviewees viewed RAS technology as simply 'a tool' to do the same operation, demanding similar skills that would be required to perform other types of minimally invasive surgery. However, others highlighted that the advanced functionality of certain robotic technology allowed operators to achieve tasks which would otherwise be impossible using other types of laparoscopic technology, often with positive patient outcomes (Beliefs about Consequences):

> "Then it actually became the tool which I thought made surgery better, so particularly something like unicompartmental arthroplasty. . .For me, it's kind of a game changer. It's an operation that is very technique dependent and it almost takes the technique out of it because it becomes a re-surfacing procedure that's computer guided, so that was the next thing." P13S, Surgeon (RAS User).

**Working in a RAS theatre: Impact on cognitive processes (Dominant domains: Memory, attention & decision processes, environmental context & resources).**   The conduct of RAS in theatre was reported to demand greater mental effort and utility of cognitive processes during early implementation. Whilst this may be similar for other innovations and technology, for RAS it was often as a result of a loss in tactile sensation whilst operating certain robotic platforms and the environmental changes introduced in theatre. Changes to the physical layout of the theatre cited included the inability to gain intraoperative access to the patient, and an increased distance between surgical team members. This was sometimes reported to create apprehension and anxiety amongst clinical staff, with associated feelings of 'losing control' (Emotion). In addition, the inability to see other clinical staff (e.g. Scrub Nurses) because the Surgeon is sometimes immersed in a RAS console provided more opportunities for distraction. Lack of mechanisms in place to accommodate for these changes could be a significant barrier to implementation:

> ". . .you're operating away from the patient, you have a distance operating role and you sometimes feel losing control and if that's the thing that bothers you too much and you're

going to be very worried, you're not going to be able to focus on the task in front of you." P12S, Surgeon (RAS User).

Other theatre changes that were presented as challenges to early implementation included the size of certain robotic platforms and the inconvenience of it occupying a large area of physical space. Some interviewees also commented that the chances of staff getting injured by RAS equipment (e.g. being hit by a moving robot arm) were higher during initial phases of implementation. One Scrub Nurse mentioned that the changes in ways of working in a RAS theatre created resistance amongst some surgical staff during early implementation:

"There's been a couple of times, a couple of surgeons have said, I can't work like this and there was one occasion where one flatly refused and said, well I can't work like this. But it was a problem we could. . . we overcame it, we did overcome it but at that point it was. . . there was resistance". P12N Scrub Nurse.

In addition, some interviewees suggested that the time it takes to set-up theatre, such as the requirement to dock certain RAS equipment and to operate with 'extra caution', meant that RAS procedures take longer to conduct during the initial phase of implementation. This prolongation could form a barrier to acceptance of RAS in the early stage of adoption.

**Social and environmental hospital structures: Access to RAS (social influences, environmental context & resources).** Scheduling issues and the inability to gain access to a robotic platform within a hospital was sometimes cited as a barrier to optimal early implementation. The scheduling issue could be particularly troublesome with the inability to move robotic equipment around the hospital to enable multi-specialty use of the equipment:

"A lot of the challenges have actually been about getting the disciplines to work, using it as a multidisciplinary tool and you can imagine the in-fighting that, you know, results then." P02S, Specialty Leader.

Another factor which was also reported to hinder access was the competitiveness amongst clinicians within hospitals, and the formation of 'elite groups' that typically comprise those who were able to gain access because of preferential treatment from management/leaders and/ or they belong to a specialty with an established reputation for conducting RAS on a regular basis. The inability to gain access to a robotic platform in order to practice RAS was also deemed to be frustrating for the Trainees:

". . .I went from a robotic job where I was on the robot every week, to a robotic job where I had a consultant that was quite early on in their trajectory who wouldn't give me the access I wanted, but the previous job gave me all the skills. And then what you create is a huge amount of frustration on everyone's part." P2SF17, Trainee.

## Late implementation barriers and enablers

**Maintenance costs (CFIR costs) & relationship with industry.** One barrier which was unique to the long-term implementation of RAS as a health service included the cost of maintaining RAS equipment, sometimes highlighted with reference to relying on sole providers to sustain service provision.

**Technological limitations of robotic platforms.** Some interviewees also acknowledged that there were technological limitations of certain robotic platforms that hindered optimal performance/use, which became more apparent during later phases of adoption:

"I think there are limitations to the technology, but I think that's because the technology is probably still in development. I think that it probably doesn't have as much utility as it can do, so I think it could be a smarter, more streamlined machine than it currently is. I think it could age things better." P2SF17, Trainee.

**Working in a RAS theatre over the longer term (Dominant domain: Memory, attention & decision processes).**   Interviewees suggested that RAS demanded higher levels of concentration and adaptation to new ways of working in theatre in the early phase of implementation. Some interviewees indicated that this becomes less of a barrier during later phases of RAS adoption because repeated practice of RAS eventually engages instinctive/automatic cognitive processes and enhances efficiency in theatre.

**Social and environmental hospital structures: Long-term access to RAS (Social influences, environmental context & resources).**   A few interviewees mentioned that some surgeons had abandoned or stalled their use of RAS at later phases of implementation. Sometimes this was cited with reference to access limitations and the complex social structures of organisations, also cited as a barrier in early implementation, as well as a lack of long-term support from management:

"One of the biggest challenges is what we would call stalling and dabbling surgeons, so they go through their learning curve, and it's not just about doing 50 or 100 procedures, but it's about doing two a week or three a week or whatever it is that is required to get them to that stage. What we see is that for the first three months, they will try their hardest and essentially stick to that routine because they've made a huge commitment with the training upfront, after three months, often, you start to see it dropping off. . ." P03I, Industry stakeholder.

The importance of management support during long-term service integration was highlighted in relation to mitigating the risks and associated cost implications of RAS underutilisation:

"But the drawback is the risk of the finance to the organisation and it's becoming a pretty objet d'art in the corner, but that you have to manage way before, during and after, but it is a risk and a drawback; you can't argue with that." P2M9, Service Manager.

**Perceived clinical, financial, and surgeon-related outcomes (Dominant domain: Beliefs about consequences, linked to knowledge).**   As in the pre-implementation phase, many interviewees acknowledged that surgical volume of RAS is important ('sweat the asset'), but also highlighted that some hospitals had reaped the benefits of RAS because they were able to maximise the use of robotic technology, resulting in (perceived) positive patient outcomes and increased cost-effectiveness. This was an enabler to longer-term implementation as internal beliefs about the benefits of RAS motivated sustained use. Impressions of RAS benefits included perceived positive patient outcomes and the ability to conduct more complex surgery with long term use:

"Well over the years we've realised that we can do more complex surgery than we used to be able to with standard minimally invasive techniques. . . So I think the complexity of the operation, it allows you to innovate, there's a lot more that you can do with robotics than you can do with standard minimally invasive surgery." P14S, Surgeon (RAS User).

However, not all interviewees shared the same beliefs. One Surgeon (RAS User) indicated that their specialty specific knowledge of the evolving evidence-base and related beliefs about the clinical outcomes of RAS discouraged their use of robotic technology in an orthopaedic setting. Likewise, the generation of evidence against the use of RAS for certain procedures impacted long-term evaluation and influenced the volume of procedures conducted over time.

As well as the clinical and cost effectiveness of RAS, interviewees also discussed the outcomes of RAS on Surgeons. The improved ergonomics achieved with RAS was reported as an enabler of long-term RAS implementation. Related to the perceived importance of maximising surgical volume, some interviewees mentioned that the continued use and resultant increase in familiarity of robotic technology can minimise stress overall in theatre, significantly reduce postural strain and enhance the longevity of a Surgeon's career:

> "... it's better ergonomics, we as a surgeon are feeling that it's going to give us more time to carry on operating like this, it's ergonomically better and comfortable for the surgeons and prolongs your careers and yes, it's much more controlled way of doing something which is very, very precise." P12S, Surgeon (RAS User).

**Perceived effects of RAS on surgical competency (Dominant domain: Skills).**   Assuming that not all surgeons have equal skill sets and capability, the ability of RAS to offer some standardisation of performance was perceived as an enabler for the longer-term implementation of RAS, particularly in a public health setting. Many interviewees highlighted that the long-term use of robotic assisted technology can reduce variability in surgical performance and 'level the playing field' amongst surgeons considered to be low/moderate performing at baseline. Although some interviewees indicated that this would not apply to 'expert' surgeons with pre-existing superior skill-sets.

**Rewards associated with RAS implementation (Dominant domain: Reinforcement).** RAS implementation was regarded as rewarding for many interviewees: this was associated with the belief that RAS adoption enables healthcare workers to build their professional profiles, become 'attractive clinicians', sometimes linked to opportunities to develop their private practice and market their affiliated organisation:

> "...and thirdly actually it's a security because if we were not to be robotic enabled, I don't think we would be as attractive an organisation as without a robot." P11S, RAS-user Surgeon.

## Supporting implementation

On the basis of the barriers and enablers identified in the interviews (Fig 1), we then sought to develop a set of potential strategies to aid implementation.

In the **pre-implementation phase**, the interview findings had suggested the main barriers/enablers, other than cost and infrastructure, were the importance of *social influence*, the role of *knowledge* and the influence of *incentivisation* (Fig 1). Early engagement of key opinion leaders–both clinical (with and without academic backgrounds) and patient champions—at the outset to support the decision to procure RAS systems is key. The effectiveness of opinion leaders was also confirmed by the behavioural evidence-base; it further highlighted that the characteristics of the most successful opinion leaders have been shown to be those who are approachable, in addition to being knowledgeable, and well connected [19–21]. Another 'Social Comparison' requirement is to obtain supporting evidence from hospitals that have successfully integrated RAS services, to demonstrate a positive impact of a new service.

Planning for downstream action for disruptive technology such as RAS should begin alongside procurement planning.

Gaps in the understanding of RAS and its impact for clinical and managerial staff requires early education prior to the procurement and arrival of the RAS system. Such education promotes a readiness to harness upcoming change (absorptive capacity) [22, 23]. Consideration should also be given to how RAS implementation can have a positive impact on attracting and retaining staff (Information about social and environmental consequences)—a hospital offering RAS procedures could be viewed as progressive, offering an environment for personal development. Any anticipated positive health-related patient outcomes from RAS should be highlighted and sits under an 'Information about health consequences' heading.

Institutional reputation can also be enhanced outside personal staff development. Suggestions that RAS implementation (by clinicians and managers) can help brand the organisation as 'innovative' and modern is not only likely to influence commissioners and attract new staff, but can incentivise a decision to implement RAS in the first place. Having such a positive backdrop promotes the initiation of RAS services into clinical practice and can be a substantial motivation for theatre and clinical staff to engage in practice change activities.

In the **early implementation phase** (including the planning for RAS installation and early use), the interview findings suggested the main barriers/enablers to effective implementation were around *social and professional role identity*, influences on *cognitive and decisional processing* for RAS theatre staff, adequately addressing the RAS *skills* gap, in addition to ongoing adaptation of the *organisational infrastructure* (Fig 1). In a different educational context to informing about RAS, early awareness and education on how RAS affects perioperative duties across theatre roles is key and should be implemented. These strategies include nursing, surgical and anaesthesia roles and focus on providing information about the social and environmental consequences of implementation. Where possible, incorporating learning from credible sources, such as personnel from other hospitals who have already implemented RAS, was seen to be particularly helpful in this regard, and recommended. Setting up internal systems from the outset, identifying lead individuals available for continuing RAS education and support can also be helpful (Social Support, Unspecified).

As for any new technology introduction, it is key to recognise and address the importance of any heightened anxiety and emotional stress for staff transitioning to RAS use. Support for staff needs in the areas of technology training, managing increased cognitive load, and any perceived increase in pressure should be incorporated early in implementation. Staff who have more experience can also provide support for newer trainees. Once selected staff are more mature in their expertise, the behavioural science literature also suggests other types of support systems may be helpful. For example, a buddy system could be implemented whereby staff are paired with nominated RAS 'experts', relevant to their respective roles (Social Support, Practical).

In addition, the development of effective internal systems for tailored RAS theatre scheduling *from the outset*—particularly if RAS is expected to be adopted across specialties in the longer term—is a suggested focus for early infrastructure adaptation (Social Support, Practical; Action planning; Problem Solving). Interview respondents repeatedly noted the lack of such a system from the outset was a major issue that hampered effective use.

In the **late implementation phase**, the interview findings had suggested that the main identified barriers/enablers affecting long term use were continuing *knowledge* gaps, the impact of ongoing *reinforcement/incentivisation* of staff, emergent impacts on *cognitive and decisional processes* in the RAS theatre, and further adaptation of the *organisational infrastructure* (Fig 1). 'Top-up' training for staff is important. The behavioural science literature suggests this type of training could usefully include the provision of feedback on RAS team performance following

observation in a simulated context (Feedback on behaviour). This could also provide a vehicle for monitoring and addressing potentially adverse RAS behaviours noted by the interviewees. Such adverse behaviours include attention deficits where staff automatically zone out/become distracted during periods when the surgeon is immersed in operating from the console (of console-based systems). Training could include the reintroduction of support mechanisms, such as the need for increased attentional/verbal prompts and cues, if deemed necessary (Prompts & Cues). Incorporating ongoing updating of knowledge about the impact of RAS on patient outcomes to RAS staff during regular top-up training would also help promote sustained use (Information about social & environmental consequences; Information about health consequences). Developing such training programmes in-house over time could also help to reduce the perceived reliance on industry for all post-procurement training.

The most efficient use of RAS was, as expected, noted to be maximised where throughput was high. Setting up internal systems for the routine collection of RAS process data *from the outset* would facilitate longer term tailoring of services to maximise throughput (Social Support, Practical). For example, if the most efficient use of RAS requires surgeons to operate 2–3 times a week, data could be monitored to ensure that this reflected in theatre schedules and implemented on a continual basis. Organisations should also consider instituting formal mechanisms (e.g. professional recognition) to incentivise staff for sustained implementation of RAS. Suggested strategies, linked to theoretical (TDF) domains, intervention functions and Behaviour Change Techniques (BCTs), across the three phases of implementation are presented in Table 2. Intervention functions, such as training and education, describe broad categories of methods used to influence behaviour, whereas BCTs comprise the 'intervention ingredients'—techniques utilised to support the delivery of an intervention [18].

## Discussion

Our study identified a range of barriers and enablers which influence the speed and effectiveness of RAS implementation at hospital sites. Our findings emphasised the dynamic and often disparate nature of the influences, highlighting that determinants of successful adoption change over time. It should be noted, however, that the research did not address value or efficacy or RAS, but only how any implementation of the technology could be optimised. Three distinct phases of implementation were evident with different characteristics in play–pre-implementation (procurement); early implementation and late implementation. It was also evident that the success of later phases of implementation are intrinsically linked to the success of the previous phases, so a consolidated action plan for the entire implementation lifecycle should be considered from the start.

The strong influence of key opinion leaders in successful implementation was evident throughout (and backed up by the evidence base) [19–21]. As part of the overall implementation of RAS involves procurement (as described in the pre-implementation phase), the relationships at this stage in the process were found to be important. There are many advantages to having engaged clinical KOLs but it was also clear that a powerful clinical opinion leader could potentially wield undue influence on a service manager (in either direction), especially if the manager lacked awareness of the wider evidence for robotic assisted surgery. Thus, whilst it is crucial to have enthusiastic and engaged clinical input, service managers should also seek to develop their own working knowledge of the RAS field, to ensure a full and balanced discussion of the case for adoption.

The findings also suggested a pivot of power and influence from clinicians to service managers over the longer implementation lifecycle. Whilst clinical influence was particularly strong in the pre-procurement phase, strong service management and organisational

**Table 2. Suggested strategies designed to optimise implementation, linked to interview themes and theoretical (TDF) domains.**

**Pre-Implementation**

| Theme(s) | Domain(s) | Intervention Function(s) | Proposed BCT(s) | Example |
|---|---|---|---|---|
| Social and Professional Roles: Impact on adoption | Social Professional Role & Identity Social Influences | Enablement | 3.2. Social Support (Practical) | Enlist support of internal or external key opinion leaders (with academic backgrounds) at the outset to promote local implementation. |
| Patient/peer pressure Group conformity Competitiveness to innovate | Social Influences | Enablement | 3.2. Social Support (Practical) 6.2. Social Comparison | Appoint clinical and patient champions to present the case for implementation. Draw attention to other hospitals with successful integrated RAS services to show benefit in comparison with existing service provision. |
| Managerial and executives influences | Social Influences Knowledge | Enablement Education | 3.2. Social Support (Practical) 5.1. Information about health consequences 5.3. Information about social and environmental consequences | Appoint clinical and patient champions to present a favourable, albeit balanced, case for implementation to managers & commissioners Inform relevant stakeholders about the potential positive consequences of RAS implementation: expected improved patient outcomes and positive impact on staff retention/recruitment. |
| Perceived Outcomes of RAS | Beliefs about Consequences Knowledge | Education | Including 5.1. and 5.3 above: 5.6. Information about emotional consequences | Outline potential positive consequences of RAS on factors such as improved staff morale. |
| Institution Branding and Profiling | Reinforcement | Incentivisation | 10.8. Incentive (outcome) | Highlight opportunities for institutional branding and identification as innovative site. |

**Set-up/Early Implementation**

| Theme | Domain(s) | Intervention Function(s) | Proposed BCT(s) | Example |
|---|---|---|---|---|
| RAS influences role modifications | Social Professional Role & Identity Social Influences | Education | 5.3. Information about social and environmental consequences 9.1. Credible Source 3.1. Social Support (Unspecified) | Raise awareness of expected changes to theatre staff roles. Present case studies of known centres who have successfully implemented RAS (e.g. real-life examples). Provide contact details of appointed individuals who can support staff to support role assimilation. |
| Competencies required to conduct RAS | Skills | Training | 4.1. Instruction on how to perform a behaviour 12.5. Adding objects to environment | Consider providing instructional cue cards (where applicable) to facilitate skill acquisition. |
| Working in a RAS theatre: Impact on cognitive processes. | Memory, Attention and Decision Processes | Training | 7.1. Prompts & Cues 3.2. Social Support (Practical) | Consider actioning verbal prompts/cues to communicate with team members if appropriate. Pair RAS staff with 'buddies' to support training development. |
| Social and environmental structures: Access to RAS | Social Influences Environmental Context & Resources | Enablement | 3.2. Social Support (Practical) 1.4. Action planning 1.2. Problem Solving | Provide an effective theatre scheduling system early in the implementation processes. Prompt regular discussions amongst staff of issues that may impede scheduled use. Encourage strategy development to overcome barriers. |

**Late Implementation**

| Theme | Domain(s) | Intervention Function(s) | Proposed BCT(s) | Example |
|---|---|---|---|---|
| Working in a RAS theatre: Automatic cognitive processes during the conduct of RAS cases | Memory, Attention & Decision Processes | Training | 2.2. Feedback on behaviour 7.1. Prompts & Cues | Evaluative feedback on team performance following observation in a simulated context. Reinstate verbal prompts and cues to combat attentional lapses if needed. |

(*Continued*)

**Table 2.** (Continued)

**Pre-Implementation**

| Theme(s) | Domain(s) | Intervention Function(s) | Proposed BCT(s) | Example |
|---|---|---|---|---|
| Social and environmental structures: Access to RAS, 'Dabbling and Stalling Surgeons' | Social Influences, Environmental Context & Resources | Enablement (with Audit and Feedback) | 3.2. Social support, practical | Follow-up multi-specialty meetings to discuss and address how to optimise theatre use, as well as to encourage greater use of RAS.<br>Present data on frequency of RAS equipment/theatre use by individual Surgeons and their respective specialties to guide effective scheduling. |
| Perceived outcomes of RAS and consideration of economic viability | Beliefs about Consequences Knowledge | Training | 5.1. Information about health consequences<br>5.3. Information about social and environmental consequences | Present regular hospital data related to RAS patient outcomes to promote understanding of (expected positive) impact on patient outcome.<br>Present hospital data related to RAS process information to aid understanding of (expected beneficial) impact on efficiency. |
| Rewards associated with RAS implementation | Reinforcement | Incentivisation | 10.6. Non-specific incentive | Identify monetary/non-monetary incentives to reward RAS activity. |

structures were key to efficient delivery over the longer term. Our findings suggested that this was often not recognised from the start, leading to delays in downstream implementation and potentially inefficient use of RAS systems. A greater focus upfront on detailed long-term planning and establishing the correct infrastructure for the longer term (rather than just planning for the procurement and early implementation) is recommended.

The dual role of industry (both as a potential enabler and barrier) was also apparent. Many cited highly positive experiences with industry especially in the process of procurement, with links to international organisations and opportunities to see the demonstrable benefits of RAS. However, the reliance on industry to sustain service provision over the longer-term was raised as a potential barrier. Similarly, a potential over-reliance on industry for the vast majority of post-procurement and long-term training was seen as problematic. As organisations mature in their RAS experience, however, reliance on industry for training may become less contentious as organisations themselves may seek to develop and cascade their own training programmes.

The impact of RAS implementation on redefinition of roles also acted in both a positive and negative way. Whilst many viewed this positively (development of new skills/roles), there were some perceived negative elements which affected implementation and buy-in, e.g. some staff feeling they had little to do in theatre post-RAS and some disquiet about surgeons potentially turning into "technicians" over the longer-term. For effective implementation over the longer-term, it is vital that such perceptions are addressed early in the implementation phase to avoid any build-up of perceived negativity with the use of the system.

## Strengths, limitations & future considerations

Our study is one of the first to collectively examine the factors that affect RAS implementation across the implementation lifecycle, and will therefore allow organisations to address barriers and enablers from the very start to the end of implementation. In addition, our novel use of a systematic behavioural approach is a strength. This methodologically innovative approach made use of a full taxonomy of strategies that have been theoretically and experimentally proven to change specific behavioural targets. This adds a level of extra rigour to our findings, such that recommendations are not solely based on experiential and "common-sense" approaches but are supported by theory informed evidence [24, 25].

Despite these strengths, there are a few weaknesses. The in-depth interview method allowed for only a relatively small interview sample, especially in view of the preponderance of RAS surgery worldwide. Also, our interview sample mainly comprised surgeons, as we were unable to secure more interviews with other RAS theatre personnel. The sampling method, although as independent as possible, was a sample primarily identified through the Royal College of Surgeons England RADAR network and through various industry connections. Although we deliberated invited interviewees that were both known to be more or less positive about RAS for balance, the natural sampling resulted in a preponderance of persons who were RAS supporters. This is worthy of further comment. The identification of any potential issues and problems with RAS outlined in this study has originated from those who are, in general, users of RAS and largely supportive, and who have substantial direct insight. In addition, the sample was largely NHS based and as such findings will likely have most resonance within the NHS settling. However, many of the issues raised were generic to RAS and thus will be applicable to both public and private healthcare sectors.

A recent article from the RCS has emphasised the need to temper the excitement surrounding surgical innovation, including RAS, and incorporate a rational evidence-based approach to practice change [26]. Interviews with participants who were less positive about RAS may have uncovered other areas worthy of consideration for implementation. Nevertheless, despite the groundswell of enthusiasm, advocations for evidence to support RAS procurement and subsequent long-term use were also communicated by stakeholders in the current study. This considered approach was also reported by industry representatives, which demonstrated the maturity of the industry interface.

Our findings suggest that the particular impact of RAS on the anaesthetist's role requires further exploration. Furthermore, our study relied on interviewees' reports, therefore, future studies could integrate complementary methods, such as direct observation, to examine the applicability of our findings.

## Conclusion

We have identified a range of barriers and enablers to the initial uptake, integration, and sustainment of RAS into clinical practice, which focused on both organisational and behavioural aspects. The influence of particular barriers and enablers was dynamic and depended on the stage of implementation. By understanding these influences, and actively planning for them in advance, these findings will aid clinicians and managers to optimise the implementation of this costly technology.

## Supporting information

**S1 File. Detailed methodological steps involved in the theory-informed implementation action plan.**
(DOCX)

## Acknowledgments

We thank the participants—specialty leads, surgeons, scrub nurses, the anaesthetist, policy-makers/commissioners and industry representatives—for their time, energy and invaluable insight to assist this research. We also thank Jared Torkington, Arul Immanuel, and Richard Kerr for providing a clinical review of the work prior to submission.

## Author Contributions

**Conceptualization:** Louisa Lawrie, Katie Gillies, David Beard, Marion K. Campbell.

**Data curation:** Louisa Lawrie.

**Formal analysis:** Louisa Lawrie, Katie Gillies, David Beard, Marion K. Campbell.

**Funding acquisition:** Katie Gillies, Loretta Davies, David Beard, Marion K. Campbell.

**Investigation:** Louisa Lawrie, Katie Gillies, David Beard, Marion K. Campbell.

**Methodology:** Louisa Lawrie, Katie Gillies, Eilidh Duncan, Loretta Davies, David Beard, Marion K. Campbell.

**Project administration:** Louisa Lawrie, Loretta Davies, David Beard, Marion K. Campbell.

**Supervision:** David Beard, Marion K. Campbell.

**Visualization:** Louisa Lawrie, Marion K. Campbell.

**Writing – original draft:** Louisa Lawrie, Katie Gillies, David Beard, Marion K. Campbell.

**Writing – review & editing:** Louisa Lawrie, Katie Gillies, Eilidh Duncan, David Beard, Marion K. Campbell.

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
