## [Decision Letter · Decision Letter 0]

14 Jun 2022

PONE-D-22-01574Barriers and enablers to the effective implementation of robotic assisted surgeryPLOS ONE

Dear Dr. Lawrie,

Thank you for submitting your manuscript to PLOS ONE. Sorry that the review for the manuscript has been inordinately delayed; a number of reviewers dropped out along the way. After careful consideration, we feel that the manuscript has merit but does not fully meet PLOS ONE’s publication criteria as it currently stands. Therefore, we invite you to submit a revised version of the manuscript that addresses the points raised below. Please submit your revised manuscript by Jul 29 2022 11:59PM. If you will need more time than this to complete your revisions, please reply to this message or contact the journal office at plosone@plos.org. Please include the following items when submitting your revised manuscript:A rebuttal letter that responds to each point raised by the academic editor and reviewer(s). You should upload this letter as a separate file labeled 'Response to Reviewers'.A marked-up copy of your manuscript that highlights changes made to the original version. You should upload this as a separate file labeled 'Revised Manuscript with Track Changes'.An unmarked version of your revised paper without tracked changes. You should upload this as a separate file labeled 'Manuscript'.

We look forward to receiving your revised manuscript.

Kind regards,

Rajagopalan Srinivasan

Academic Editor

PLOS ONE

Journal Requirements:

2. Please include an ethics statement in your Methods section, and include the name of the IRB, the approval number, the type of consent obtained, and a statement as to whether the IRB approved the consent procedures.

Reviewers' comments:

Reviewer's Responses to Questions

**Comments to the Author**

1. Is the manuscript technically sound, and do the data support the conclusions?

Reviewer #1: Yes

2. Has the statistical analysis been performed appropriately and rigorously? 

Reviewer #1: N/A

3. Have the authors made all data underlying the findings in their manuscript fully available?

Reviewer #1: Yes

4. Is the manuscript presented in an intelligible fashion and written in standard English?

Reviewer #1: Yes

5. Review Comments to the Author

Reviewer #1: Overall, this is a well written and well conducted piece of research. It is interesting to read and important. I think it should be published. I have some relatively superficial concerns about the presentation of the implementation strategy especially, and a few small suggestions to improve the document/reporting, but overall I think this is a strong piece of work.

I am not an expert in the qualitative methods or theoretical frameworks and the paper probably needs a different reviewer for these aspects.

There is little information about the identification and recruitment of sites and participants, apologies if I have missed this. I appreciate you purposively sampled, but where from and how were these participating sites and individuals selected? Did you advertise the study, use industry or professional organisations to identify units? And how did you identify people to interview? Can you be confident that you have the negative views of those less inclined to RAS or just enthusiasts?

Was the tendency to recruit from the NHS intentional? I would imagine there are many private sites with robots and issues may differ?

Recruitment & RADAR specifically is first mentioned (without spelling out the acronym) in the last paragraph of the discussion – clearly this needs to be addressed much earlier in the methods.

I feel if you are going to say that your implementation plan is ‘evidence-based’, it implies to me that you have based your plan on the whole body of the literature – is there an underlying or preceding systematic review that has helped inform this, or is your ‘evidence-based’ implementation strategy just based on your evidence? (there could, of course, be conflicting evidence in the literature). I think if it is just based on the evidence produced from this study alone, I think you should drop the ‘evidence-based’ and just say you have produced an RAS implementation plan.

My view on this is strengthened by the implementation strategy itself, which concludes that the use of key opinion leaders is important in implementation – and seems to recommend the use of eminence above evidence and anecdotal evidence (for example, from other sites) rather than high-quality research. I feel this is an implementation strategy based on an uncritical evaluation of the data (ie a recommendation based on the experience of interviewees, rather than what should be done). If you are going to produce recommendations, I personally think you should take a wider societal perspective on the value of evidence-based decision making in healthcare & choosing wisely, especially if the cost-effectiveness of unproven technologies is considered. Either that, or be very clear what you have produced and its purpose.

Is the implementation strategy a set of recommendations from the study group, or a set of tips and tricks to get RAS implemented based on the experience of those who have succeeded, which may or may not have been good for patients or public-funded healthcare more widely? I’m not sure I feel this has been critically considered in the paper, although it is recognised in parts the decisions were made for reasons other than evidence of better patient outcomes. It is half addressed in the discussion, but in an uncritical way and the strengths and weaknesses of this evidence in terms of it being produced from a ‘pro-robot’ community is important to consider. I would be relatively concerned by some of the findings if I was reading this as a senior healthcare commissioner.

Abstract: ‘The acceptance that RAS is a future technology’ doesn’t quite make sense (and does imply a certain bias in your views towards RAS) – could you re-word please?

6. PLOS authors have the option to publish the peer review history of their article (what does this mean?). If published, this will include your full peer review and any attached files.

Reviewer #1: No

---

## [Author Response · Author response to Decision Letter 0]

13 Jul 2022

Apologies, but I can no longer locate the financial disclosure section on the submission site that we need to complete. I have pasted the relevant information below. This is also in the manuscript:

MKC and DB were funded via an unrestricted grant (https://www.intuitive.com/en-gb) awarded by Intuitive Surgical (European Research Board). The funder had no role in the conception, design, conduct, analysis, or interpretation of the study.

Please see attached cover letter for our responses to the review presented in a more readable (table) format. We have also copied and pasted the information below.

We would like to thank the Editor and the Reviewer for feedback on our manuscript. We believe we have sufficiently addressed all comments provided – please see below for our responses to all queries/comments. 

Thank you for considering the manuscript for publication. We look forward to hearing from you. 

Yours sincerely,

Dr Louisa Lawrie

Reviewer #1 

Overall, this is a well written and well conducted piece of research. It is interesting to read and important. I think it should be published. I have some relatively superficial concerns about the presentation of the implementation strategy especially, and a few small suggestions to improve the document/reporting, but overall I think this is a strong piece of work.

RESPONSE: We would like to thank the reviewer for their positive feedback.

There is little information about the identification and recruitment of sites and participants, apologies if I have missed this. I appreciate you purposively sampled, but where from and how were these participating sites and individuals selected? Did you advertise the study, use industry or professional organisations to identify units? And how did you identify people to interview? Can you be confident that you have the negative views of those less inclined to RAS or just enthusiasts?

RESPONSE: Thank you for pointing this out. We have now included more information regarding our recruitment processes within the Data Collection sub-heading of the Methods section. We have also been more transparent about the limitations of our sampling method within the Discussion section of the paper.

Was the tendency to recruit from the NHS intentional? I would imagine there are many private sites with robots and issues may differ?

RESPONSE: The project was linked to a wider piece of work (REINFORCE) supported by the NHIR HS&DR funding scheme, which is NHS based. The emphasis on NHS exploration was probably a result of this although we did not deliberately target NHS groups. With respect to private providers, which we agree were not well represented in the surveys, there may have been some minor differences in barriers/enablers but the majority of findings are likely to be transferable. Overall, we would probably suggest that our results are more likely applicable to the NHS and have stated this as a potential limitation. 

Recruitment & RADAR specifically is first mentioned (without spelling out the acronym) in the last paragraph of the discussion – clearly this needs to be addressed much earlier in the methods.

RESPONSE: We have now included this information within the Data Collection section of the paper. 

I feel if you are going to say that your implementation plan is ‘evidence-based’, it implies to me that you have based your plan on the whole body of the literature – is there an underlying or preceding systematic review that has helped inform this, or is your ‘evidence-based’ implementation strategy just based on your evidence? (there could, of course, be conflicting evidence in the literature). I think if it is just based on the evidence produced from this study alone, I think you should drop the ‘evidence-based’ and just say you have produced an RAS implementation plan.

RESPONSE: This is a reasonable point. We appreciate the term ‘evidence-based’ used in this way can be confusing, and so we have removed any references to an ‘evidence-based implementation plan’. We have instead included the following sentence in the study aims for clarification: “Provide an optimised theory-informed plan for effective RAS implementation, with the incorporation of evidence-based implementation strategies that have been adapted to address the barriers/enablers that affect RAS service adoption.” We hope this clarifies the purpose as the reviewer has kindly suggested.

My view on this is strengthened by the implementation strategy itself, which concludes that the use of key opinion leaders is important in implementation – and seems to recommend the use of eminence above evidence and anecdotal evidence (for example, from other sites) rather than high-quality research. I feel this is an implementation strategy based on an uncritical evaluation of the data (ie a recommendation based on the experience of interviewees, rather than what should be done). If you are going to produce recommendations, I personally think you should take a wider societal perspective on the value of evidence-based decision making in healthcare & choosing wisely, especially if the cost-effectiveness of unproven technologies is considered. Either that, or be very clear what you have produced and its purpose. 

RESPONSE: We would like to clarify that our implementation plan is ‘evidence-based’ because it incorporates techniques established in the field of implementation science that have been proven to effectively initiate change and facilitate the implementation of interventions. The techniques that we have included within this plan have been further tailored to address the specific barriers (and optimise the enablers) that were identified by our interviewees as affecting the uptake of RAS. Some of them were not feasible to address within the scope of this project (e.g. the cost of the equipment), and so we focussed on aspects that hospitals could more readily influence (e.g. such as the use of dedicated RAS champions to facilitate role transitions). 

Is the implementation strategy a set of recommendations from the study group, or a set of tips and tricks to get RAS implemented based on the experience of those who have succeeded, which may or may not have been good for patients or public-funded healthcare more widely? I’m not sure I feel this has been critically considered in the paper, although it is recognised in parts the decisions were made for reasons other than evidence of better patient outcomes. It is half addressed in the discussion, but in an uncritical way and the strengths and weaknesses of this evidence in terms of it being produced from a ‘pro-robot’ community is important to consider. I would be relatively concerned by some of the findings if I was reading this as a senior healthcare commissioner.

RESPONSE: The implementation plan/strategies that we have outlined in the paper include techniques that have been proven to be effective (in the field of behavioural/implementation science) for introducing interventions into the healthcare system. These techniques have been adapted to address the barriers and optimise the enablers that were identified by our interviewees. Our implementation plan is therefore, in part, based on the reports of those who have had experience of RAS implementation, but it is not based on the recommendations that they have provided. Rather, our implementation plan is based on their reports of the barriers and enablers of RAS implementation, and the evidence-base with regards to the most effective techniques that are designed to address these barriers/enablers. Our methods section details the steps that we have taken to design this implementation plan, which includes reference to a systematic mapping process whereby barriers/enablers are categorised according to the framework that we used (the Theoretical Domains Framework, TDF), and then mapped to effective techniques designed to initiate change. We have included more detail about the steps involved in the design of our implementation plan. This has been submitted as a supplementary file (referenced in the Methods section), as opposed to the main body of the text for those interested in the technicalities of our approach. We opted to provide a lay summary without behavioural science terminology within the manuscript.

We have also more critically reflected on our sampling method within the Discussion section. The reviewer is correct that ultimately the sample was imbalanced towards “pro-robot”, but we made firm attempts to keep some balance in the sampling by questioning several “anti-robot” personnel.

Abstract: ‘The acceptance that RAS is a future technology’ doesn’t quite make sense (and does imply a certain bias in your views towards RAS) – could you re-word please?

RESPONSE: Thank you for pointing this out. The sentence has now been reworded to read “…views around the uptake and expansion of RAS in the future.” We would like to clarify that this is a sentence which describes the content of participant responses from the interviews, as opposed to our personal views.

---

## [Decision Letter · Decision Letter 1]

15 Aug 2022

Barriers and enablers to the effective implementation of robotic assisted surgery

PONE-D-22-01574R1

Dear Dr. Lawrie,

We’re pleased to inform you that your manuscript has been judged scientifically suitable for publication and will be formally accepted for publication once it meets all outstanding technical requirements.

Kind regards,

Rajagopalan Srinivasan

Academic Editor

PLOS ONE

Reviewers' comments:

Reviewer's Responses to Questions

**Comments to the Author**

1. If the authors have adequately addressed your comments raised in a previous round of review and you feel that this manuscript is now acceptable for publication, you may indicate that here to bypass the “Comments to the Author” section, enter your conflict of interest statement in the “Confidential to Editor” section, and submit your "Accept" recommendation.

Reviewer #1: All comments have been addressed

2. Is the manuscript technically sound, and do the data support the conclusions?

Reviewer #1: Yes

3. Has the statistical analysis been performed appropriately and rigorously? 

Reviewer #1: Yes

4. Have the authors made all data underlying the findings in their manuscript fully available?

Reviewer #1: Yes

5. Is the manuscript presented in an intelligible fashion and written in standard English?

Reviewer #1: Yes

6. Review Comments to the Author

Reviewer #1: Thank you for these revisions and responses which have improved the manuscript. The paper is interesting, reads well and I think it is suitable for publication.

7. PLOS authors have the option to publish the peer review history of their article (what does this mean?). If published, this will include your full peer review and any attached files.

Reviewer #1: **Yes: **Andrew Metcalfe

---

## [Editor Report · Acceptance letter]

18 Aug 2022

PONE-D-22-01574R1 

Barriers and enablers to the effective implementation of robotic assisted surgery 

Dear Dr. Lawrie:

I'm pleased to inform you that your manuscript has been deemed suitable for publication in PLOS ONE. Congratulations! Your manuscript is now with our production department. 

Kind regards, 

on behalf of

Dr. Rajagopalan Srinivasan 

Academic Editor

PLOS ONE